# Learning Constraints from Offline Dataset via Inverse Dual Values Estimation

## Abstract

To develop safe control strategies, Inverse Constrained Reinforcement learning (ICRL) infers constraints from expert demonstrations and trains policy models under these constraints. Classical ICRL algorithms typically adopt an *online* learning diagram that permits boundless exploration in an interactive environment. However, in realistic applications, iteratively collecting experiences from the environment is dangerous and expensive, especially for safe-critical control tasks. To address this challenge, in this work, we present a novel Inverse Dual Values Estimation (IDVE) framework. To enable *offline* ICRL, IDVE dynamically combines the conservative estimation inherent in offline RL and the data-driven inference in inverse RL, thereby effectively learning constraints from limited data. Specifically, IDVE derives the dual values functions for both rewards and costs, estimating their values in a bi-level optimization problem based on the offline dataset. To derive a practical IDVE algorithm for offline constraint inference, we introduce the method of 1) handling unknown transitions, 2) scaling to continuous environments, and 3) controlling the degree of sparsity regularization. Under these advancements, empirical studies demonstrate that IDVE outperforms other baselines in terms of accurately recovering the constraints and adapting to high-dimensional environments with diverse reward configurations.

## 1 Introduction

In order to deploy Reinforcement Learning (RL) algorithms to solve safety-critical applications, the control policy must conform to some underlying constraints in the environment (Liu et al., 2021). However, in many real-world tasks, due to the inherent complexity of environmental dynamics, the optimal constraint is often time-varying, context-sensitive, and rooted in the human experience. These constraints are difficult to specify manually with prior knowledge and may not be readily available to RL agents in policy learning.

To resolve these problems, Inverse Constraint Reinforcement Learning (ICRL) considers inferring constraints from the expert demonstrations. Existing ICRL algorithms (Scobee & Sastry, 2020; Malik et al., 2021; Liu & Zhu, 2022; Gaurav et al., 2023; Liu et al., 2023; Papadimitriou et al., 2023) follow an online learning paradigm where the agent can explore and collect experience from an interactive environment. However, the boundless exploration often involves unsafe behaviors that potentially violate the underlying constraints. *Such a shortcoming is fundamental* since the violation of constraint contradicts the primary goal of safe control and may cause significant loss in practical applications, especially for the algorithms that require a large number of training samples.

An effective method to overcome the above limitations is designing an offline ICRL algorithm that relies only on offline datasets for constraint inference. This task is challenging, primarily due to 1) offline datasets can cover only a partial knowledge of the training environment and the algorithm must learn to handle the lack of knowledge in unvisited states. 2) without actively exploring specific actions and observing their outcomes, the offline ICRL algorithm must accurately identify the unsafe behaviors by relying only on the offline dataset.

To address the aforementioned challenges in offline ICRL, in this work, we propose an Inverse Dual Value Estimation (IDVE) framework. IDVE reformulates constraint inference as a regularized policy learning problem, thereby ensuring both the safety and conservatism of a control strategy. This is achieved by regularizing the deviation from the expert policy and incentivizing the agent to operate within the distribution of offline datasets. By leveraging the Lagrange duality, we deduce an

analytic solution for our problem by embedding the optimal visitation distribution into value functions of costs and rewards. Such functions, being sensitive to the agent's performance in reward maximization and cost avoidance, provide an effective quantification of the agent's behavioral feasibility. To learn these value functions from datasets, we derive a bi-level optimization objective that alternatively updates the value functions for rewards and costs in an offline diagram.

For practical usage, we design a IDVE algorithm that can effectively handle the incomplete or unknown transition, scale to a continuous environment, and control the sparsity of learned constraints. These advancements enable IDVE to significantly outperform other baselines by inferring more accurate constraints and safer control policies. We conduct an in-depth evaluation to study its performance under various hyperparameters and how well the constraints transfer to new environments.

Our *main contributions* are as follows: 1) To the best of our knowledge, there is no prior solution for offline ICRL. As the first attempt, our IDVE framework derives dual value functions from regularized policy learning and proposes the bi-level optimization method for updating value functions, which could serve as a strong cornerstone for future advancements. 2) We design an IDVE algorithm that solves critical problems caused by the continuous environment, unknown transitions, and scale of constraints. These advancements bridge IDVE to accomplish practical applications.

## 2 RELATED WORK

**Inverse Constrained Reinforcement Learning (ICRL).** Prior ICRL methods extend the maximum entropy framework (Ziebart et al., 2008) for learned constraints from both the expert demonstrations and the interactive MDP environment (without the constraints). In the discrete state-action space, some recent research (Scobee & Sastry, 2020; McPherson et al., 2021) inferred the constrained sets for recording infeasible state-action pairs in Constrained MDP, but these studies were restricted to the environments with known dynamics. A subsequent work (Malik et al., 2021) extended this approach to continuous state-action spaces with unknown transition models by utilizing neural networks to approximate constraints. To enable constraint inference in stochastic environments, (Papadimitriou et al., 2023) inferred probability distributions over constraints by utilizing the Bayesian framework, and (Baert et al., 2023) incorporate the maximum causal entropy objective (Ziebart et al., 2010) into ICRL. Some recent works explore ICRL under different settings, e.g., (Gaurav et al., 2023) extended ICRL to infer soft constraints, and (Liu & Zhu, 2022) explored ICRL under the multi-agent setting. Striving for efficient comparisons, (Liu et al., 2023) established an ICRL benchmark across various RL domains. However, these algorithms primarily target online ICRL that infer constraints by interacting with environments instead of with only offline datasets.

**Offline Reinforcement Learning.** Offline RL utilizes a data-driven RL paradigm where the agent learns the control policy exclusively from static datasets of previously collected experiences (Levine et al., 2020). To mitigate the distributional shift between training samples and testing data, previous offline RL solutions commonly involve constraining the learned policy to the data-collecting policy (Fujimoto et al., 2019; Kumar et al., 2019), making conservative estimates of future rewards (Kumar et al., 2020; Yu et al., 2021), and developing uncertainty-aware action selector (Janner et al., 2019; Kidambi et al., 2020). Some recent advancements on Offline RL (Sikchi et al., 2023; Xu et al., 2023) studied a regularized policy optimization problem with convex objective and linear constraints. A recent IRL work (Yue et al., 2023) considered recovering conservative rewards from offline datasets, but none of these methods has studied the offline constraint inference.

## 3 PROBLEM FORMULATION

**Constrained Reinforcement Learning (CRL).** A CRL problem is commonly based on a stationary Constrained Markov Decision Process (CMDP) $\mathcal{M} \cup c := (\mathcal{S}, \mathcal{A}, p_{\mathcal{T}}, r, c, \epsilon, \rho_0, \gamma)$ where: 1) $\mathcal{S}$ and $\mathcal{A}$ denote the space of states and actions. 2) $p_{\mathcal{T}} \in \Delta_{\mathcal{S} \times \mathcal{A}}^{\mathcal{S}}$[1] defines the transition distributions. 3) $r : \mathcal{S} \times \mathcal{A} \to [0, R_{\max}]$ and $c : \mathcal{S} \times \mathcal{A} \to [0, C_{\max}]$ denotes the reward and cost functions. 4) $\epsilon \in \mathbb{R}_+$ denotes the bound of cumulative costs. 5) $\rho_0 \in \Delta^{\mathcal{S}}$ denotes the initial states distribution. 6) $\gamma \in [0, 1)$ is the discount factor. The goal of CRL policy $\pi \in \Delta_{\mathcal{S}}^{\mathcal{A}}$ is to maximize the expected discounted rewards under known constraints:

$$\arg\max_{\pi} \mathbb{E}_{p_{\mathcal{T}}, \pi, \rho_0} \Big[ \sum_{t=0}^{T} \gamma^t r(s_t, a_t) \Big] \text{ s.t. } \mathbb{E}_{p_{\mathcal{T}}, \pi, \rho_0} \Big[ \sum_{t=0}^{T} \gamma^t c(s_t, a_t) \Big] \le \epsilon \quad (1)$$

---

[1]$\Delta^{\mathcal{X}}$ denotes the probabilistic simplex in the space $\mathcal{X}$, and $\Delta_{\mathcal{Y}}^{\mathcal{X}}$ denotes a function maps $\mathcal{Y}$ to $\Delta^{\mathcal{X}}$.

Following the setting in Malik et al. (2021), we are mainly interested in the hard constraints such that $\epsilon = 0$. Striving for clarity, we define the CMDP with the known cost as $\mathcal{M} \cup c$, and the CMDP without cost (i.e., CMDP$\backslash c$) as $\mathcal{M}$. Accordingly, the visitation distribution $d^\pi \in \Delta^{\mathcal{S} \times \mathcal{A}}$ (i.e., the normalized occupancy measure) produced by policy $\pi$ can be denoted as:

$$d^\pi(s,a) = (1-\gamma)\pi(a|s) \sum_{t=0}^{\infty} \gamma^t p(s_t = s|\pi) \tag{2}$$

where $p(s_t = s|\pi)$ defines the probability of arriving state $s$ at time step $t$ by performing policy $\pi$.

**Inverse Constraint Reinforcement Learning.** Note that traditional CRL problems often assume the constraint signals $c(\cdot)$ are directly observable from the environment, but in real-world problems, instead of observing the constraint signals, we often have access to expert demonstrations $\mathcal{D}_E$ that adhere to these constraints, and the agent is required to recover the constraint models from the dataset. This task is challenging because various combinations of rewards and constraints can explain the same expert demonstrations. Striving for the identifiability of solutions, ICRL algorithms Malik et al. (2021); Liu et al. (2023); Papadimitriou et al. (2023) typically assume that *reward signals are observable and the goal is to recover only the constraints*, in contrast to Inverse Reinforcement Learning (IRL) Ziebart et al. (2008), which aims to learn rewards from an unconstrained MDP.

**Identifiability Issue.** Similar to IRL, the optimal constraint in ICRL is not uniquely identifiable. indicating that multiple constraints may equivalently explain the expert behaviors. To address this issue, ICRL algorithms aim to learn the *minimal constraints* under which the imitation agent can reproduce the behaviors of the expert (Scobee & Sastry, 2020). These constraints are defined so as to *prohibit risky movements that could yield cumulative rewards exceeding those obtained by the expert*. This is because we assume that experts optimally maximize rewards within their constraints. Hence, if an agent surpasses an expert's rewards, it indicates inherent risk in the move.

**From Online to Offline ICRL.** Classic ICRL algorithm typically follows an online learning paradigm where the agent iteratively collects experience by interacting with the environment and using that experience for updating constraints and policy. Nevertheless, in many realistic settings, online interaction is impractical, either because data collection is expensive (e.g., in robotics, educational agents, or healthcare) or dangerous (e.g., in autonomous driving, or healthcare). To extend ICRL to the offline setting, we formally define the problem of *offline ICRL* as follows:

**Definition 1.** *(Offline ICRL) Let $\mathcal{D}^E = \{s_n^E, a_n^E, r_n^E\}_{n=1}^{N_E}$ denote the expert dataset generated by the agent adhering to the unobserved ground-truth constraints. Let $\mathcal{D}^{\neg E} = \{s_n, a_n, r_n\}_{n=1}^{N_{\neg E}}$ denote the sub-optimal dataset generated by the agent without knowing the ground-truth constraints. Given an offline dataset $\mathcal{D}^O = \{\mathcal{D}^E, \mathcal{D}^{\neg E}\}$ and the threshold $\hat{\epsilon}$, an offline ICRL problem requires estimating the cost function $\hat{c}(\cdot)$ such that the reward-maximising policy $\hat{\pi}$ learned under the inferred constraint can reproduce expert demonstration $\mathcal{D}_E$.*

The challenge of solving an offline ICRL problem lies in the absence of an MDP $\mathcal{M}$ that the algorithm can interact with, more specifically,

- To infer the correct constraint, traditional online ICRL algorithms rely on active exploration of the environment for identifying the unsafe trajectories that yield larger cumulative rewards compared to expert ones. However, offline ICRL algorithms have no access to the environment.
- The demonstration dataset $\mathcal{D}_o$ captures only the partial information of the environment, and thus the offline ICRL algorithms must learn a conservative constraint and policy representation, thereby mitigating the influence of epistemic uncertainty due to the incomplete knowledge.

## 4 CONSTRAINT INFERENCE VIA DUAL REINFORCEMENT LEARNING

The offline forward constraint-solving function is defined by the regularized policy learning objective. We use $d^O$ to represent the visitation distribution in the offline dataset $\mathcal{D}^O$, and $D_f(\cdot \,||\, \cdot)$ denotes the $f$-divergence between two distributions. Instead of maximizing the reward, we augment the reward with a divergence regularizer to prevent it from deviating beyond the coverage of the offline data. This guarantees that the agent adheres to a conservative policy. Specifically, we aim to maximize $J(\pi) = \mathbb{E}_{d^\pi(s,a)}[r(s,a)] - \xi_r D_f(d^\pi(s,a) \,||\, d^O(s,a))$, i.e.

$$\max_{\pi} \mathbb{E}_{d^\pi(s,a)}[r(s,a)] - \xi_r D_f(d^\pi(s,a) \,||\, d^O(s,a)) \ \ s.t. \ \ d^\pi(s,a)c(s,a) \leq \epsilon \ \forall s,a \tag{3}$$

Motivated by the computational efficiency, and inspired by Sikchi et al. (2023), we reformulate the aforementioned objective (3) into a convex problem. Specifically, we aim to identify a visitation distribution that adheres to the Bellman-flow constraints:

$$\max_{d(s,a)c(s,a)\leq 0, d(s,a)\geq 0} \mathbb{E}_d[r(s,a)] - \xi_r D_f(d \,||\, d^O) \tag{4}$$

$$\text{s.t. } \sum_{a\in\mathcal{A}} d(s,a) = (1-\gamma)d_0(s) + \gamma \sum_{(s',a')} d(s',a')p(s|s',a'), \ \forall s\in\mathcal{S}$$

To derive a solution for the problem (4), we introduce the dual variables $V^r$ to consider the Lagrangian dual problem. Please find the detailed derivation in appendix B.1.

$$\min_{V^r}\max_d \ \mathbb{E}_d[\delta_V^r] - \xi_r D_f(d \,||\, d^O) + (1-\gamma)\mathbb{E}_{d_0}[V^r] \ \text{ s.t. } \ d(s,a)c(s,a) \leq \epsilon \text{ and } d(s,a)\geq 0 \tag{5}$$

*where $\delta_V^r(s,a) = r(s,a) + \sum_{s'\in\mathcal{S}} p_{\mathcal{T}}(s'|s,a)\gamma V^r(s') - V^r(s)$.*

Intuitively, $\delta_V^r(s,a)$ defines the *temporal difference error in policy evaluation* and the *advantages in policy improvement* respectively. *In this study, we focus on hard constraints, denoted by $\epsilon = 0$. This implies that the feasible set is constrained by the condition $d(s,a)c(s,a) \leq 0$ for all $s,a$. We assume $c(s,a)$ is derived from the state function $\mathbf{c}(s')$, indicating state safety, with $c(s,a) = \mathbb{E}_{s'\sim P(s'|s,a)}[\mathbf{c}(s')]$. We provide the closed-form solution for the inner optimization problem in Equation (5) when $\epsilon = 0$. Note that a similar derivation under the non-constraint case can be found in Lee et al. (2021), Sikchi et al. (2023).*

**Proposition 1.** *Assume the following: 1) The learned visitation distribution $d(s,a) > 0$ for all $(s,a)$ such that $d^O(s,a) > 0$. 2) The range of the derivative of the function $f$, i.e., $f'$, includes 0. In other words, there exists some $x \geq 0$ s.t. $f'(x) = 0$.*

*The optimal solution for the inner optimization problem in 5, denoted as $d^*$, is given by :*

$$d^*(s,a) = d^O(s,a)\mathbb{1}_{c(s,a)=0}w_f^*\left(\delta_V^r(s,a)\right) \tag{6}$$

*Substituting $d^*$ into problem 5, the problem becomes:*

$$\min_{V^r} \ \mathbb{E}_{d^O}\left[\xi_r \mathbb{1}_{\delta_V^c(s,a)=0}f_p^*\left(\frac{\delta_V^r(s,a)}{\xi_r}\right)\right] + (1-\gamma)\mathbb{E}_{d_0}[V^r] \tag{7}$$

*where $w_f^*$ and $f_p^*$ is related to the convex function $f$ specified in f-divergence: $w_f^*(y) = \max(0, f'^{-1}(\frac{y}{\xi_r})), f_p^*(y) = \mathbb{E}_{w^*}[y] - f(w^*(y))$.*

*The proof is in Appendix B.2. Following Proposition 1, we observe that the inner optimization problem projects every unsafe $\omega_f^*(\delta_V^r(s,a))$ to $d^*(s,a) = 0$. Now, considering the problem of inversely learning the cost $c(s,a)$ from expert demonstration $d^E$, the following proposition provides conditions for $c(s,a)$ to solve the inverse problem, ensuring that $d^E$ is the solution to problem (3).*

**Definition 2.** *The set of optimal visitation distributions is defined as those distributions that satisfy Bellman flow constraints and achieve a higher cumulative reward with a regularizer, denoted as $O = \{d : J(d) > J(d^E)\} \cap \{d : \sum_{a\in\mathcal{A}} d(s,a) = (1-\gamma)d_0(s) + \gamma\sum_{(s',a')} d(s',a')p(s|s',a'), \ \forall s\in\mathcal{S}\}$.*

**Proposition 2.** *The $c(s,a)$ is a feasible solution for the inverse constraint learning problem if and only if 1) For every $d \in O$, there exists at least one $c(s,a) > 0$ when $d(s,a) > 0$ and 2) $c(s,a) = 0$ for all $d^E(s,a) > 0$.*

*Motivated by this proposition, we formalize our inverse constraint learning problems as a bi-level optimization problem. Initially, we solve the forward constraint equation using the formula 5, which can be viewed as sampling from the optimal visitation distribution $O$. Once the solution $V^r$ is obtained, representing the optimal solution under a learned cost, we update the cost function to ensure $c(s,a) > 0$ for some (s,a) such that $d^r(s,a) > 0$, derived from $d^r(s,a) = d^O(s,a)w_f^*(\delta_V^r(s,a))$. This iterative process is repeated until convergence. We define the objective function for the bi-level optimization problem as follows:*

$$\min_{V^r} \ \mathbb{E}_{d^O}\left[\xi_r \mathbb{1}_{\delta_V^c(s,a)\leq 0}f_p^*\left(\frac{\delta_V^r(s,a)}{\xi_r}\right)\right] + (1-\gamma)\mathbb{E}_{d_0}[V^r] \tag{8}$$

$$\max_{V^c} \ \sum_{(s,a)\in T^e} w_f^*(\delta_V^r(s,a) - (\delta_V^c(s,a))_+) - \sum_{(s,a)\in T^v} w_f^*(\delta_V^r(s,a) - (\delta_V^c(s,a))_+) \tag{9}$$

*where $\delta_V^c(s,a) = \sum_{p_\mathcal{T}(s'|s,a)} \gamma V^c(s') - V^c(s)$ and $V^c(s) = \mathbb{E}_{\pi,p_\mathcal{T},\rho_0}[\sum_t \gamma^t c(s_t, a_t)|s_0 = s]$ defines the cost value function. We use $(x)_+$ to represent $\max(x, 0)$. The state-action set visited by the expert is denoted as $T^e = (s, a) : d^E(s, a) > 0$, and the set not visited by the expert is denoted as $T^v = (s, a) : d^E(s, a) = 0$. In this formula: 1. We use the advantage of the cost function $\delta_V^c(s, a)$ to represent the cost function $c(s, a)$, aligning with our setting where $c(s, a) = \sum_{p_\mathcal{T}(s'|s,a)} \gamma V^c(s') - V^c(s)$ should solely depend on the future state and not the action. 2. When using the gradient method to update 9, we maximize $\delta_V^c$ to 0 when $d^E(s, a) > 0$. This ensures that $d^E(s, a) > 0$ leads to $\delta_V^c = 0$. Furthermore, by minimizing $\delta_V^c$ to 0 where $d^E(s, a) = 0$, $V^c$ will have a gradient only when $\omega_f^*(\delta_V^r) > 0$. This guarantees an increase in the cost value in the state-action pair where $d^r(s, a) > 0$. We can prove that, following such a learning process, the optimizing process can maintain the learned optimal occupancy $d^*$ in each round as a better policy than $d^E$ with respect to $J(d)$. See appendix B.4.*

### 4.1 ANALYSIS OF THE IDVE

A critical challenge in ICRL is finding the minimum constraint that can explain the behavior of expert agents (Scobee & Sastry, 2020; Malik et al., 2021; Liu et al., 2023). Developing a sparse constraint is critical for fostering generalizability. Without sparsity, one can learn an excessively restrictive constraint that blocks all the movement not covered by the expert dataset. While such a constraint may accurately reflect the observed expert behaviors, it lacks practical utility because: 1) the expert dataset might not record all possible movements, and 2) it cannot be generalized to environments with even minor modifications to dynamics (see experiments in Section 6.2), which commonly appears in bridging the Sim-to-Real gap in practice. To achieve this goal, it is important to encourage the sparsity of constraints. We claim that IDVE achieves sparsity by identifying only those constraint-violating movements that yield high rewards, aligning with the goals of ICRL (see Section 3).

*As an illustrative example, we consider chi-square divergence for $D_f$:*

$$f = (x-1)^2 \text{ and } f'(x) = 2(x-1) \text{ and } f'^{-1}(x) = 1 + \frac{x}{2} \tag{10}$$

$$w_f^*(x) = \mathbb{1}(x > -2\xi_r)\left(1 + \frac{x}{2\xi_r}\right) = \left(1 + \frac{x}{2\xi_r}\right)_+ \tag{11}$$

*In the cost learning step, if the temporal difference $\delta_V^r(s, a) \leq -2\xi_r$ for any $(s, a)$ pair, indicating negligible future rewards for the action in that state, our objective ensures $\nabla w_f^*(\cdot) = 0$ in the cost learning objective (equation (9)). Consequently, the cost-value function $V^c(s)$ remains unchanged in the low-reward state-action pairs, which encourages the sparsity of updates in $V^c(s)$. Intuitively, this method makes the cost function only sensitive to regions exhibiting constraint-violating behaviors, particularly those associated with rewards exceeding the expert level. In the forward solving step, for any state-action transition $(s, a, s')$ with nonzero cost $c(s, a)$, $\mathbb{1}_{\delta_V^c \leq 0}$ will block the update of $V^r(s)$ from $V^r(s')$. This acts as a safe Bellman operator, defining the optimal value function $V^*(s)$ as $V^{safe}(s) = \max_{a \in \mathcal{A} \wedge c(s,a) \neq 0}\left(r(s, a) + \gamma P(s'|s, a)V^{safe}(s')\right)$.*

**An example of the sparse constraint.** We take the gridworld as an example. If $\xi_r = \frac{1}{2}$, as shown in Figure 1, for $(s, a, s') = ([4, 2], \text{'move up'}, [4, 1])$, since $V^r([4, 2]) - V^r([4, 1]) < 0$ and $r([4, 1], \text{"move up"}) = -1$, we can conclude that $\delta_V^r([4, 2], \text{"move up"}) = V^r([4, 2]) - V^r([4, 1]) + r([4, 1], \text{"move up"}) < -1$. Therefore, $\delta_V^r - \max(\delta_V^c, 0)$ will be clipped by the indicator $\mathbb{1}(x > 2\xi_r)$ during the computation of $w^*(x)$ in objective (9). Thus, the value of $V^c([4, 2])$ and $V^c([4, 1])$ remains unchanged.

However, for $([4, 2], \text{'move down'}, [4, 3])$, $\delta_V^r([4, 2], \text{"move down"}) = V^r([4, 3]) - V^r([4, 2]) + r([4, 2], \text{"move down"}) > -1$, so

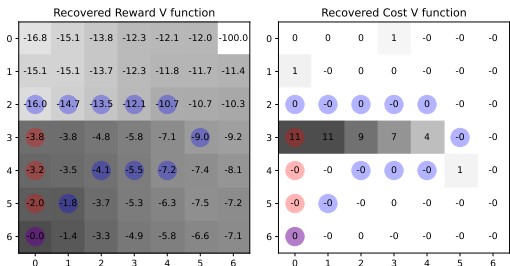

Figure 1: An example of recovery $V^r$ and $V^c$ under the experiment setting 1 in Section 6.1. $[x, y]$ denotes the state, and the action is the moving direction (e.g., move right).

the gradient in $\delta_V^c$ will increase the value of $V^c([4,3])$, marking $[4,3]$ as an unsafe space. In the next optimizing forward problem, $[4,3]$ will be blocked from back-propagating the gradient to any other $V^r$, to ensure a safe policy improvement.

# 5 PRACTICAL IMPLEMENTATION

In this section, we introduce the practical implementation of IDVE (see Algorithm 1) by proposing the following key updates to our IDVE objective (8) and (9). We use $\chi$-square divergence as $D_f$.

## 5.1 REWRITING OF FORWARD OPTIMIZATION PROBLEM

*Following Sikchi et al. (2023), we simplify the bi-level optimization objective (8) by replace $\xi_r$ and $\gamma$ with temperature parameter $\lambda$ and sparsity parameter $\alpha$. becomes:*

$$\max_{V^c} \sum_{(s,a)\in T^e} (\delta_V^r(s,a) - (\delta_V^c(s,a))_+ - \alpha)_+ - \sum_{(s,a)\in T^v} (\delta_V^r(s,a) - (\delta_V^c(s,a))_+ - \alpha)_+ \quad (12)$$

$$\min_{V^r} \lambda \mathbb{E}_{d^O}\left[\mathbb{1}_{\delta_V^c(s,a)\leq 0} f_p^*(\delta_V^r(s,a))\right] + (1-\lambda)\mathbb{E}_{d_0}[V^r] \quad (13)$$

*Intuitively, $\lambda$ governs the level of conservatism in optimization, representing the tradeoff between maximizing immediate rewards (first term) and aligning with offline data (second term). On the other hand, $\alpha$ dictates the lower bound for clipping and update equations, capturing the trade-off associated with the sparsity of constraint recovery.*

## 5.2 SCALING TO CONTINUOUS ENVIRONMENT

*Sampling from $d^E(s,a) = 0$ is challenging due to limited expert trajectory samples, leaving some $(s,a)$ pairs absent. We address this by sampling $(s,a)$ from $d^*(s,a)$ whenever $(s,a^e)$ exists in expert trajectory samples. we can store the state-action pairs that significantly deviate from expert behavior in the replay buffer $B^v$. To implement this, we use a Gaussian representation for the actor-network $\pi_\psi \sim \mathcal{N}(\mu,\sigma)$ to extract the policy from learned $V^r$ and $Q^r$. The maximization of log likelihood under optimal state-action visitation is expressed as: $\max_\psi \mathbb{E}_{s,a\sim d^*}[\log \pi_\psi(s,a)] = \max_\psi \mathbb{E}_{s,a\sim d^O}\left[\mathbb{1}_{\delta_V^c(s,a)\leq 0}\omega_f^*(\delta_V^r(s,a))\log \pi_\psi(s,a)\right]$ ($\delta_V^c$ and $\delta_V^c$ denote advantages in policy update). We optimize:*

$$\max_{V^c} \mathbb{E}_{d^E}[(\delta_V^r - \max(\delta_V^c, 0))] - \alpha)_+ - \mathbb{E}_{d^v}[(\delta_V^r - \max(\delta_V^c, 0))] - \alpha)_+ \quad (14)$$

Since the actions in the violation buffer result in higher rewards but are also more likely to lead to unsafe states, the reduction of $w_f^*(\delta_V^r(s,a) - \max(\delta_V^c(s,a), 0)$ in $d^v$ effectively increases the values of $V^c(s')$ for states followed by state actions pair $(s,a)$ stored in the violation action replay buffer.

## 5.3 TACKLING UNKNOWN TRANSITIONS IN THE OFFLINE LEARNING

In our IDVE algorithm, the computing of $\delta_V^r(s,a)$ and $\delta_V^c(s,a)$ function requires complete knowledge of the transition $(s,a,r,s')$. In value update, online learning algorithms can interact with the environment to explore the resulting states $s'$ by performing an action $a$ on the state $s$. However, in the offline setting, the dataset might not cover the returns of performing a specific action, and $s'$ becomes unavailable without the interactive environment. To circumvent this issue, we define:

$$\delta_V^r(s,a) = Q^r(s,a) - V^r(s) \text{ and } \delta_V^c(s,a) = Q^c(s,a) - V^c(s) \quad (15)$$

Since the reward signals are known in the offline ICRL dataset, we adopt a semi-gradient update rule to update $Q^r$. Specifically, we fix $V^r$ and update $Q^r$ by:

$$\min_{\phi^r} \mathbb{E}_{(s,a,s')\sim\mathcal{D}}\left[(Q_{\phi^r}^r(s,a) - (r(s,a) + \gamma V_{\theta^r}^r(s')))^2\right] \quad (16)$$

where $\phi^r$ and $\theta^r$ denote the parameters of Q and V functions. *For cost value learning, we update $Q^c$ using the equation in 12 and approximate $V^c(s')$ with the maximum possible learned cost Q function to propagate high learned cost in violation action to an unsafe state in the offline dataset, i.e., $V^c(s') = \max_{(s,a,s')\in\mathcal{D}^O}[\gamma Q_{\theta^c}^c(s,a)]$. This update rule ensures the consistency of $c(s,a)$ across different $(s,a)$ pairs that lead to same destination $s'$*

# 6 EMPIRICAL RESULTS.

**Running Settings.** By following Malkin et al. (2022), we adopt the following evaluation metric: 1) *Constraint Violation Rate*, which assesses the likelihood of a policy violating a constraint in a given

---

**Algorithm 1:** Inverse Dual Values Estimation (IDVE)

---

**Require:** Offline dataset $\mathcal{D}^O = \{\mathcal{D}^E, \mathcal{D}^{\neg E}\}$, Running iterations $I$;

1: Initialize $Q_{\phi^c}^c = 0$, $V_{\theta^c}^c = 0$;

2: Run offline RL to warm up $Q_{\phi^r}^r$, $V_{\theta^r}^r$, $\pi_\psi$, and the violating action replay buffer $B_v = \{\emptyset\}$;

3: **for** $i = 1 \ldots I$ iterations **do**

4:     Sample violate action $a_v$ from $a_v \sim \mathbb{E}_{s \sim \mathcal{D}}[\pi_\psi(s)]$, add $(s, a_v)$ to buffer $B_v$;

5:     Update $Q_{\phi^r}^r$ by minimizing the TD error (16) with dataset $\mathcal{D}^O$;

6:     Update $V_{\theta^r}^r$ with the dual value objective (13) and dataset $\mathcal{D}^O$;

7:     Update $Q_{\phi^c}^c$ by minimizing the behavioral gap to experts (objective (14)) with dataset

8:     Update $V_{\theta^c}^c$ with the objective $\min_{\phi^c} \mathbb{E}_{(s,a,s') \sim \mathcal{D}^O} \left[ \min(\gamma V_{\theta^c}^c(s') - Q_{\phi^c}^c(s,a), 0) \right]$;

9:     Update $\pi_\psi$ with $\max_\psi \mathbb{E}_{s,a \sim d^O} \left[ \mathbb{1}_{\delta_V^c(s,a) \leq 0} \omega_f^*(\delta_V^r(s,a)) \log \pi_\psi(s,a) \right]$;

10: **end for**

---

trajectory, and 2) *Feasible Cumulative Rewards*, which calculates the total rewards accumulated by the agent before violating any constraints. 3) the *success rate* of reaching the destination for the grid-world environment. We run experiments with 5 different seeds and present the mean $\pm$ std results for each algorithm. Appendix A.3 reports the detailed settings and random seeds.

**Comparison Methods.** Due to the lack of offline ICRL baselines, we mainly compare IDVE with its variants and other relevant offline control methods, including 1) **IDVE w/o S** removes the control of sparsity by removing the clipping term in function (12) 2) **IDVE w/o A** excludes the violating action buffer $B^v$ from objective (9) by following only the expert density for learning cost values. 3) **Offline IL** follows the *Inverse Soft Q-Learning* Garg et al. (2021) method that infers reward value functions from offline data to imitate expert policy. 4) **Offline RL** refers to the recently proposed *f-DVL* Sikchi et al. (2023) algorithm that leverages the dual and offline reward function to control.

## 6.1 DISCRETE ENVIRONMENT

We utilize grid-world environments for evaluating our algorithm. These environments consist of 7x7 discrete maps, each designed with four unique constraint map settings (Figure 2). Within these environments, each agent is permitted to perform eight actions: moving up, down, right, left, or in one of four diagonal directions. The primary objective for every agent is to navigate from the start to the end point,

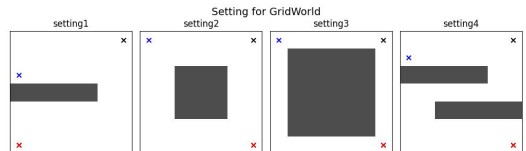

Figure 2: Four settings in grid-world. Blue, red, and black mark the starting, target, and absorbing states. The algorithms should infer the constrained region (gray) from demonstrations.

taking the shortest possible route while avoiding specific constrained states. The agent receives a -1 reward for each step taken until it reaches its destination. To enable offline ICRL, we provide an expert dataset $\mathcal{D}^E$ and a sub-optimal dataset $\mathcal{D}^{\neg E}$ (collected random walks) for each environment. The size of the offline dataset $\mathcal{D}^O = \{\mathcal{D}^E, \mathcal{D}^{\neg E}\}$ is 50 (trajectories) (Check details in Appendix A.1).

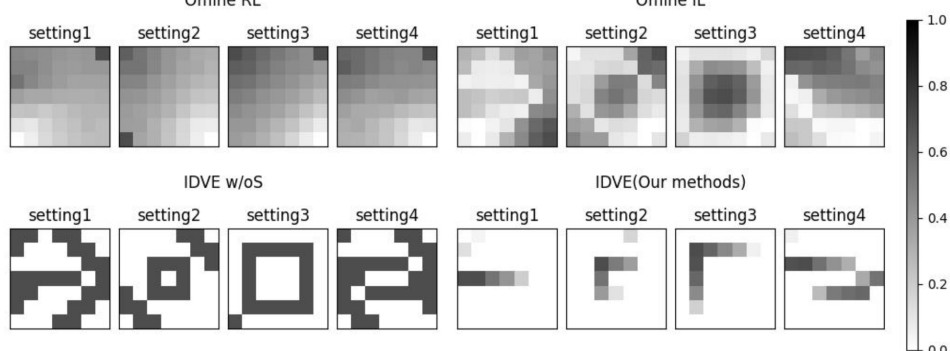

Figure 3: The visualization for the recovered constraints in four settings.

**Constraint Visualization.** Figure 3 visualizes the normalized value functions learned by different methods. By comparing with the ground-truth constraints in Figure 2, we find our

IDVE successfully identifies the most sparse constraint, which plays a critical role in enabling the agent to navigate safely to its intended destination. An important phenomenon is that the learned cost values of IDVE $w/oS$ significantly become denser, thereby constraining lots of safe regions. It reflects the necessity of applying a sparsity regularizer. The value function learned by the imitation model fails to capture the accurate constraint.

**Sensitivity to hyper-parameters.** We conduct an in-depth study to investigate how the key parameters ($\lambda$ and $\alpha$) in IDVE influence the performance of constraint recovery. Figure 4 visualizes the results. By scaling the regularizing $\alpha$ from 0 to $-\infty$ ( $-\infty$ indicates no regularization), we observe an increase in constraint density, which demonstrates the efficacy of IDVE in controlling sparsity. When we elevate $\lambda$ from 0.4 to 0.55, IDVE shows a bias towards behaviors that maximize cumulative rewards (refer to our objective 13). Consequently, the constraints that prevent agents from reaching the target states in the fewest steps diminish in significance. We find this phenomenon is most apparent when $\alpha = 0$, and it becomes less apparent as the density of the constraint increases ($\alpha$ becomes larger).

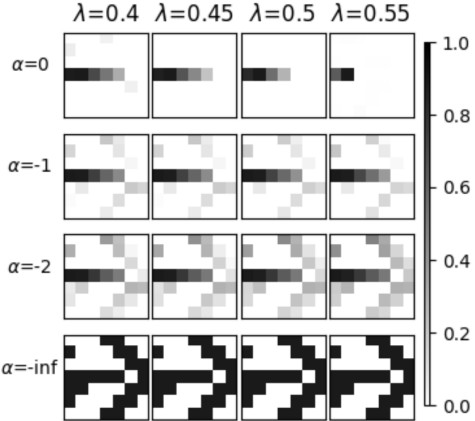

Figure 4: Effect of hyper-parameters to Cost Value function $V^c$.

## 6.2 GENERALIZABILITY TO NEW ENVIRONMENTS.

We study how well the inferred constraints can be generalized to new environments with *different starting and target states*. In the experiment, the constraints are learned with data collected for training environments (Figure2), and these constraints are evaluated under the new environment in Figure 5.

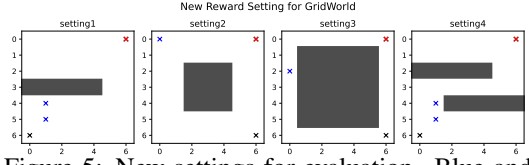

Figure 5: New settings for evaluation. Blue and red denote the new starting and target states.

Table 1 shows the results. The results show IDVE can outperform other baselines as it uniquely achieves a high success rate while maintaining a low cost. In contrast, IDVE $w/oS$ does not achieve comparable performance, as over-dense constraints prohibit actions that have not been explored by the expert, yet may not necessarily be infeasible. The policy learned by offline RL also does not scale well, and we omit the offline IL as the distribution of rewards changes when the environment is modified, rendering the rewards learned by IL invalid.

Table 1: Generalization performance in a Grid-World Environment. We bold the largest rewards corresponding to safe policy without constraint violation.

| Expert demonstration | Env | Offline RL | | | IDVEw/o S | | | IDVE | | |
|---|---|---|---|---|---|---|---|---|---|---|
| | | Reward | Success Rate | Cost | Reward | Success Rate | Cost | Reward | Success Rate | Cost |
| 10% expert | setting1 | -5.5 | 100% | 100% | $-\infty$ | 0% | **0%** | $-\infty$ | 80% | 40% |
| | setting2 | -5.0 | 100% | 100% | $-\infty$ | 0% | **0%** | **-6.0** | 100% | **0%** |
| | setting3 | -6.0 | 100% | 100% | $-\infty$ | 60% | **0%** | -6.6 | 100% | 100% |
| | setting4 | -5.0 | 100% | 100% | $-\infty$ | 0% | **0%** | -7.5 | 100% | 60% |
| 50% expert | setting1 | -5.5 | 100% | 100% | $-\infty$ | 0% | **0%** | **-7.1** | 100% | **0%** |
| | setting2 | -5.0 | 100% | 100% | $-\infty$ | 0% | **0%** | **-6.0** | 100% | **0%** |
| | setting3 | -6.0 | 100% | 100% | $-\infty$ | 80% | **0%** | $-\infty$ | 80% | 100% |
| | setting4 | -5.0 | 100% | 100% | $-\infty$ | 0% | **0%** | **-7.1** | 100% | **0%** |
| 100% expert | setting1 | -5.5 | 100% | 100% | $-\infty$ | 0% | **0%** | **-7.5** | 100% | **0%** |
| | setting2 | -5.0 | 100% | 100% | $-\infty$ | 0% | **0%** | **-6.0** | 100% | **0%** |
| | setting3 | -6.0 | 100% | 100% | **-6.0** | 100% | **0%** | **-6.0** | 100% | **0%** |
| | setting4 | -5.0 | 100% | 100% | $-\infty$ | 0% | **0%** | **-7.0** | 100% | **0%** |

## 6.3 CONTINUOUS ENVIRONMENT

Our Continuous environments utilize MuJoCo Todorov et al. (2012), a virtual simulator suited for robotic control tasks. To extend MuJoCo for constraint inference, we modify the MuJoCo environments by incorporating predefined constraints into each environment. We design constraints from

different perspectives for these agents: 1) We draw inspiration from two experiments conducted in Liu et al. (2023), where robots are forbidden from moving backward when it's easier to move backward than to move forward (e.g., Half-Cheetah and Walker). We also create two other constraints inspired by real-world experiments. First, we impose a constraint on the agent's maximum forward speed to simulate real-world speed limits. In the second environment, we enforce a constraint on the agent's leg angles to prevent movement of its first leg. Table 3 summarizes the environment settings.

The results and corresponding learning curve can be found in Table 2 and Figure 6. Across all environments, both IDVE and IDVE w/oS exhibit robust performance in imitating high-performing policies offline while maintaining the safety of the policy. *This finding aligns with our expectations*, as the sparsity regularizer is primarily tailored for enhancing generalizability (Section 4.1). Since MuJoCo uses identical training and testing environments, the benefits of encouraging sparsity are not readily apparent. When it comes to offline IL, it achieves a low cost when operated in offline mode, but it demonstrates relatively inferior reward-maximizing performance. Meanwhile, for offline RL, the associated costs rise substantially since it is not sensitive to the underlying constraint. Figure 7 in the appendix also visualizes the constraints of the Blocked Half-cheetah. The recovered rewards of offline IL show that its reward function assigns negative rewards to the unsafe states. Conversely, our method yields a sparse cost function centered around point 0, which effectively facilitates safe movements by solely discouraging backward movement from the start point.

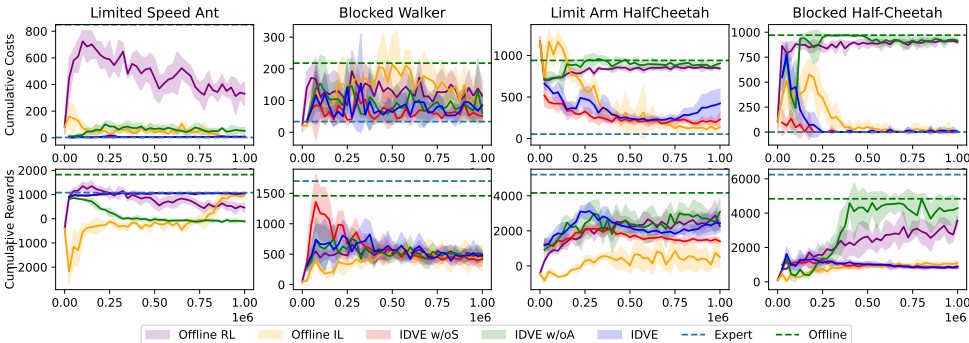

Figure 6: The cumulative rewards and the costs from the evaluation during training.

Table 2: MuJoCo testing performance. We report the average cumulative rewards and cumulative costs in 10 runs. The best average performance is highlighted in bold.

| Method | Limited Speed Ant | Limit Arm HalfCheetah | Blocked Walker | Blocked Half-Cheetah |
|---|---|---|---|---|
| **Cumulative Rewards** | | | | |
| Offline RL | 461.10±182.45 | 2,269.86±198.45 | 495.62±60.49 | 3,565.56±234.58 |
| Offline IL | **1,284.85±77.07** | 850.74±374.66 | 440.13±90.62 | 726.65±61.68 |
| IDVE w/oA | -108.67±53.34 | **3,099.46±670.42** | 458.47±79.86 | **4,297.64±571.42** |
| IDVE w/oS | 1,043.50±12.07 | 1,405.13±133.72 | 419.28±94.90 | 828.01±78.04 |
| IDVE | 1,061.11±21.90 | 2,433.99±445.43 | **483.97±38.21** | 901.62±74.09 |
| **Cumulative Costs** | | | | |
| Offline RL | 330.24±79.84 | 330.24±79.84 | 115.32±42.67 | 902.10±14.19 |
| Offline IL | 6.06±2.97 | **133.56±48.47** | 70.04±39.16 | 10.30±20.60 |
| IDVE w/oA | 52.36±24.02 | 905.60±38.22 | 60.76±24.84 | 917.04±28.89 |
| IDVE w/oS | **4.40±0.41** | 228.97±69.93 | **51.38±11.31** | **0.00±0.00** |
| IDVE | 9.48±3.75 | 421.50±157.57 | 85.26±36.10 | 5.04±10.08 |

# 7 CONCLUSION

In this paper, we present a IDVE framework as the first attempt to facilitate offline ICRL. This is achieved by deriving dual value functions from regularized policy learning and formulating a bi-level optimization problem to update these value functions. To enhance practical applicability, we introduce a IDVE algorithm that effectively addresses unknown transitions, continuous environments, and insufficient sparsity. Empirical results demonstrate the performance of IDVE in various settings. A promising avenue for future research involves extending our method to accommodate diverse ICRL configurations, such as soft constraints in stochastic environments.

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

## A  IMPLEMENTATION AND ENVIRONMENT DETAILS

### A.1  DISCRETE ENVIRONMENTS

Our discrete environments use a 7x7 gridworld map, with four different settings for constraint learning. In each setting, we assign two absorbing states: one as the reward state in the training environment and another as the reward state in the transfer learning test. We employ value iteration to obtain the expert demonstration, and we shape the reward as $\hat{r} = r - 100c$ by subtracting a cost*100 term to ensure that the demonstration is safe. For offline data collection, we sampled from random walks starting at grids [1,1], [1,5], [5,1], and [5,5]. To enrich the dataset with additional information, the agent was instructed to choose a state it had not visited previously whenever making a move.

To obtain the value function recovered by inverse imitation learning, we employed inverse q learning. All the recovered cost value functions in Figure 3 are used with a single set of hyperparameters.

### A.2  CONTINUOUS ENVIRONMENTS

Our virtual environments are based on Mujoco. We provide more details about the virtual environments as follows:

1. **Blocked HalfCheetah and Blocked Walker** These two environments are taken from Liu et al. (2023). In these environments, The agent controls a robot that can move faster backward than forward. The reward is based on the distance it moves between the current and previous time steps, along with a penalty related to the magnitude of the input action. We have defined a constraint that restricts movement in the region where the X-coordinate is less than or equal to 0, allowing the robot to move forward only.

2. **Limit Arm HalfCheetah**

   In this environment, the agent controls a robot to move forward and obtain rewards related to moving distance. However, we enforce a constraint on the agent's leg angles to prevent the use of its first leg. The cumulative sum of the angles at the agent's front thigh, shin, and foot joints (represented as $\sum_t |\theta_{fthigh,t}| + |\theta_{fshin,t}| + |\theta_{ffoot,t}|$) must not exceed 100 degrees.

3. **Limited Speed Ant**

   In this environment, the agent controls a robot to move forward and obtain rewards related to moving distance. However, it must not exceed a speed limit of 0.2, resulting in obtaining fewer rewards compared to no constraint.

Table 3: Summary of Environment Settings

| Environment | Constraints |
|---|---|
| Blocked Half-cheetah | X-Coordinate $> 0$ |
| Blocked Walker | X-Coordinate $> 0$ |
| Limited Speed Ant | X-velocity $< 0.2$ |
| Limit Arm HalfCheetah | $\sum_t |\theta_{fthigh,t}| + |\theta_{fshin,t}| + |\theta_{ffoot,t}| < 100$ |

**Data Generation.** To obtain offline data, we adhere to the guidelines provided in Fu et al. (2020). We utilize the data from a 100000-step early stop SAC replay buffer as an offline dataset. Additionally, we sample 100 trajectories from the expert dataset. When sampling from the offline dataset, we maintain a balanced approach by selecting half of the samples from the offline dataset and half from the expert trajectories. We evaluate the learned policy every 25,000 steps and run a total of 1 million training steps.

A.3 IMPLEMENTATION DETAILS

We implement the violation buffer as a queue. We sample actions, denoted as $a_v$, from the actor network given a state denoted as $s_e$, from the pair $(s_e, a_e)$ in expert demonstrations. We calculate the difference between the Q-values of these two actions in the reward Q-function, i.e., $Q^r(s_e, a_v) - Q^r(s_e, a_e)$, and select the top K actions with the highest differences to add to the buffer. When the buffer reaches its capacity, we remove actions that have lower values in terms of $Q^r(s_e, a_v) - V^r(s_e)$.

To ensure stable trainning, we have added L1 regularization to both the Q and value functions. Across our set of environments, we maintain the same hyperparameters (except for the temperature parameter $\lambda$, where we use 0.7 in the Walker environment and 0.8 in others). We utilize a two-layer MLP with 256 hidden units and a ReLU activation function. The complete set of hyperparameters used in our experiments is presented in Table 4 and the details of dataset used are list in Table 5.

| Hyperparameter | Value |
|---|---|
| Batch Size | 256 |
| Violate Action Buffer Size | 10000 |
| Violate Action Sample Size k | 500 |
| Policy Learning Rate | 3e-4 |
| Value Learning Rate | 3e-4 |
| LR Decay Schedule | Linear |
| L1 Regularization Coefficient | 0.001 |
| Clipping Coefficient $\alpha$ | -10 |
| Random Seed $\alpha$ | 1 2 3 4 5 |

Table 4: Hyperparameters for our experiments.

Table 5: Details of offline dataset and expert demonstration

| Task Name | Number of Offline Transitions | Number of Expert Transitions |
|---|---|---|
| Limited Speed Ant | 100000 | 100000 |
| Limit Arm HalfCheetah | 100000 | 100000 |
| Blocked Walker | 100000 | 9985 |
| Blocked Half-Cheetah | 100000 | 100000 |

### A.4 EXPERIMENTAL EQUIPMENT AND INFRASTRUCTURES

We ran the experiment on a cluster that has multiple RTX 3090 GPUs, each with 24 GB of memory. There is only one running node. With the aforementioned resources, running one seed in the virtual environment takes about 40 minutes.

## B PROOF AND DERIVATION

### B.1 DERIVING LAGRANGIAN OF PROBLEM (4)

By introducing the Lagrange multiplier $V^r$ and $\lambda_d$. The Lagrangian of problem (4) can be:

$$\mathbb{E}_d[r(s,a)] - \xi_r D_f(d \,||\, d^O) - V^r(s)(\sum_{a \in \mathcal{A}} d(s,a) - (1-\gamma)d_0(s) - \gamma \sum_{(s',a')} d(s',a')p(s|s',a'))$$

$$=\mathbb{E}_d\left[r(s,a) + \gamma \sum_{(s',a')} d(s',a')p(s|s',a'))V^r(s') - V^r(s)\right] + (1-\gamma)\mathbb{E}_{d_0}[V^r] - \xi_r D_f(d \,||\, d^O)$$

$$=\mathbb{E}_d[\delta_V^r] - \xi_r D_f(d \,||\, d^O) + (1-\gamma)\mathbb{E}_{d_0}[V^r]$$

### B.2 PROOF OF PROPOSITION 1

*Proof.* In order to solve:

$$\max_d \; \mathbb{E}_d[\delta_V^r - \delta_V^c] - \xi_r D_f(d \,||\, d^O) + (1-\gamma)\mathbb{E}_{d_0}[V^r] \tag{17}$$

$$\text{s.t. } d(s,a)c(s,a) \leq 0 \text{ and } d(s,a) \geq 0 \,\forall s, a \tag{18}$$

Assume $d(s,a) > 0$ implies $d^O(s,a) > 0$. We introduce a new variable $w(s,a) = \frac{d(s,a)}{d^O(s,a)}$ and redefine our optimization objective as follows:

$$\max_w \; \mathbb{E}_{d^O}[w(s,a)\delta_V^r - \xi_r f(w(s,a))] + (1-\gamma)\mathbb{E}_{d_0}[V^r(s)] \tag{19}$$

$$\text{s.t. } \omega(s,a)c(s,a) \leq 0 \text{ and } \omega(s,a) \geq 0 \tag{20}$$

Without the constraint on $d(s,a)$, the maximization of the first part is equivalent to finding the convex conjugate of the function $f$, denoted as $f^*(y) = \max_x(xy - f(y))$. However, we must consider the non-negativity constraint $d(s,a) \geq 0$ and feasible constraint $\omega(s,a)c(s,a) \leq 0$.

We can form the Lagrangian function:

$$\mathcal{L}(V^r, \lambda^c, \lambda) = \mathbb{E}_{d^O}[w(s,a)\delta_V^r - \xi_r f(w(s,a))] + (1-\gamma)\mathbb{E}_{d_0}[V^r(s)] + \sum_{s,a}(\lambda(s,a)\omega(s,a) - \lambda_c(s,a)c(s,a)\omega(s,a))$$

The KKT condition of this problem is:

1.Primal feasibility $\omega(s,a)c(s,a) \leq 0, \omega^*(s,a) \geq 0 \;\forall s, a$

2.Dual feasibility $\lambda_c^*(s,a) \geq 0, \lambda^*(s,a) \geq 0 \ \forall s,a$

3.Stationarity $\frac{\partial \mathcal{L}}{\partial V^r} = d^O(s,a)(-f'(\omega^*(s,a)) + \delta_V^r(s,a) + \lambda^*(s,a) - \lambda_c^*(s,a)c(s,a)) = 0 \ \forall s,a$

4.Complementary Slackness $\omega^*(s,a)\lambda^*(s,a) = 0, \lambda_c^*(s,a)\omega^*(s,a)c(s,a) = 0$

Using Complementary Slackness, we have two case:

Case 1:$\lambda_c^*(s,a) = 0$, which means solution without introducing constraint is feasible. Following a similar proof given in Sikchi et al. (2023)(Refer to equation 43 in appendix of the cited paper for more details), we solving the optimal solution with stationary equation and first complementary slackness equation:

$$\omega^*(s,a) = \omega^r(s,a) = \max(0, f'^{-1}(\frac{\delta_V^r(s,a)}{\xi_r}))$$

. and $\omega^r(s,a)$ satisfies primal feasibility $\omega^*(s,a)c(s,a) = 0$.

Case2:$\lambda_c^*(s,a) \neq 0$, which means $\omega^*(s,a)c(s,a) = 0$. Which means we need solve following question. Note that such solution is possible if only there exists some x s.t. $f'(x) = 0$.

$$\omega^*(s,a) = \max(0, f'^{-1}(\frac{\delta_V^r(s,a) - \lambda_c^*(s,a)c(s,a)}{\xi_r})) \tag{21}$$

$$\omega^*(s,a)c(s,a) = 0 \tag{22}$$

Summarizing these two case and replace $\omega^*$ with $d^*$, we have

$$d^*(s,a) = d^O(s,a)\mathbb{1}_{c(s,a)=0} \max(0, f'^{-1}(\frac{\delta_V^r(s,a)}{\xi_r})) \tag{23}$$

$\square$

### B.3 PROOF OF PROPOSITION 2

*Proof.* "→": If $d = d^E$ solve the problem. then the $c(s,a) = 0$ for all $d^E(s,a) > 0$, otherwise the $d^E$ is not feasible. And if there exist a $d \in O$ s.t. $d(s,a)c(s,a) = 0 \ \forall s,a$, then $d$ is feasible and $d$ is optimal than $d^r$ in primal problem, which leads to constridiction.

"←": If $\forall d \in O$ there is at least one $(s,a)$,s.t. $dc \neq 0$, then all $d \in O$ are all not feasible. Since $c(s,a) = 0 \ \forall (s,a)s.t.d^E(s,a) > 0 \ d^E$ is feasible, so $d^E$ is optimal.

$\square$

### B.4 PROPOSITION OF MAINTAINING OPTIMALITY

**Proposition 3.** *For finite state space $\mathcal{S}$ and finite action space $\mathcal{A}$. If $d$ is the only optimal solution under constraint. We initialize $c(s,a) = 0$, then $V_{(0)}^r = \arg\min_{V^r} \mathbb{E}_{d^O}\left[\xi_r f_p^{(\frac{\delta_V^r(s,a)}{\xi_r})}\right] + (1 - \gamma)\mathbb{E}_{d_0}[V^r]$, i.e., the optimal solution without constraint. By alternatively solving for $V^r$ and if resulting distribution is not $d^E$, updating $c$ to project at least one $w_f^*(\delta_V^r(s,a))$ to 0, while keeping $c(s,a) = 0$ for all $d^E(s,a) > 0$.*

*Then, the algorithm will converge to $d^E$. During the learning process, the visting distribution $d_{(k)}^*$ corresponding to $V_{(k)}^*$ maintain $J(d_{(k)}^*) \geq J(d^E)$.*

*Proof.* We know $J(d) \geq J(d^E)$ since $d^E$ is always a candidate for a solution in each round of the update. We now prove that if $d_{(k)}^* \neq d^E$, we can always find some $s,a$ such that $d_{(k)}^*(s,a) > 0$ and $d^E(s,a) > 0$. First, we observe that for any policy $d$ that visits fewer state-action pairs than $d^E$, i.e., if $\{(s,a) : d(s,a) > 0\} \subset \{(s,a) : d^E(s,a) > 0\}$, we have $J(d) < J(d^E)$. Otherwise,

$d$ is feasible, which would contradict the uniqueness of optimality of $d^E$. Thus, we can always find some $(s, a)$ in $\{(s, a) : d(s, a) > 0\}$ but not in $\{(s, a) : d^E(s, a) > 0\}$.

From this, we also know that if the algorithm converges, it will always converge to $d^E$. We notice that for every round of updating, the number of zeros in $c_{(k)}(s, a)$ is increasing. Since in every round $k$, the optimal visitation distribution $d^*_{(k)}(s, a)$ with respect to $V^*_{(k)}$ satisfies $d^*_{(k)}(s, a)c_{(k-1)}(s, a) = 0$, and in round $k$, we must set some $c_{(k)}(s, a)$ to be above 0 for $d^*_{(k)}(s, a) > 0$.

Now, the spaces $\mathcal{S}$ and $\mathcal{A}$ are finite. We know the algorithm will converge to $d^E$ since there are only a finite number of combinations of $(s, a)$. $\qquad\square$

## C    EXPERIMENTS

This section contains additional results.

### C.1    RESULTS IN GRID WORLD

| Metric | Violation Rate | Cumulative Return | Success Rate |
|---|---|---|---|
| Offline RL | 100% | **-6** | 100% |
| Offline Imiataion learning | **0%** | -11.0 | 100% |
| IDVE | **0%** | -11.0 | 100% |
| IDVE w/o Sparse | **0%** | -11.0 | 100% |

Table 6: Comparison of Learned policy

### C.2    RESULTS IN MUJOCO

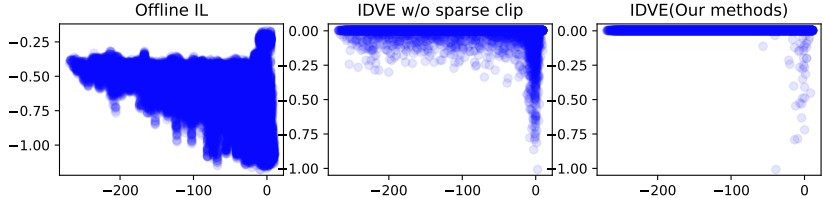

Figure 7: Recovered reward/cost results in Blocked HalfCheetah. The x-axis represents the x-position of the agent, and the y-axis represents the learned normalized reward/cost.

### C.3    INFLUENCE OF HYPERPARAMETERS ON TRANSFER LEARNING IN GRIDWORLD

We are comparing the impact of hyperparameters on both the successful transfer rate and the constraint violation rate under different parameter combinations. As depicted in the figures 8 and 9, the success rate and violation rate improve with an increase in the number of expert demonstrations. Additionally, when the clip coefficient $\alpha$ is set to a higher value, it becomes more frequent to reach the new reward state but also more likely to violate the constraint.

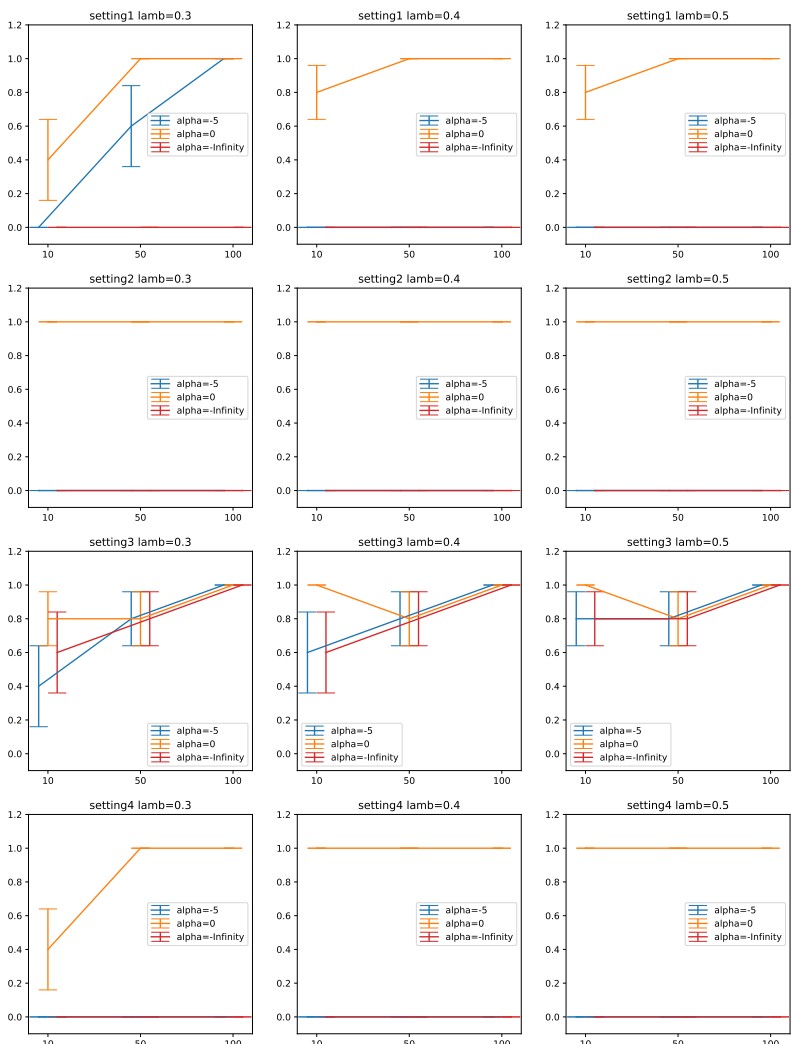

Figure 8: The x-axis represents the percentage of provided expert data, and the y-axis represents the probability of trained agents reaching the reward state.

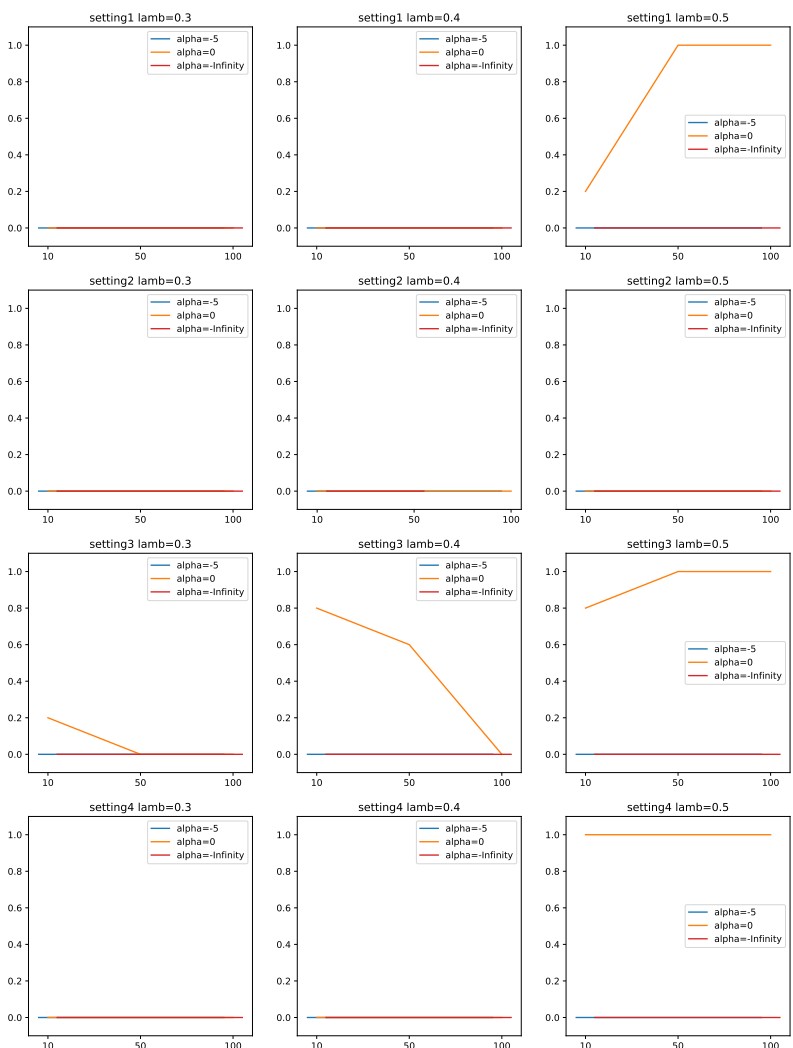

Figure 9: The x-axis represents the percentage of provided expert data, and the y-axis represents the probability of trained agents violating the constraint.

