# OpenReview forum: "Learning Constraints from Offline Dataset via Inverse Dual Values Estimation"
_ICLR.cc/2024/Conference — Submitted to ICLR 2024_

### Official Review · Reviewer_KCBL · 2023-10-28

**Soundness:** 2 fair
**Presentation:** 2 fair
**Contribution:** 2 fair
**Rating:** 3
**Confidence:** 4

**Summary:**

This paper proposes an inverse constraint learning RL problem based on the agreement with expert demonstrations. The authors introduce a strange and potentially problematic way to define the cost, i.e., as the divergence to the expert occupancy measure ($D_f(d\|d^E)-\epsilon$). The paper then constructs a DICE framework and then models this problem as a bi-level optimization problem: at one level, solving an offline RL problem and on the other level, minimizing the divergence of the optimal state-action occupancy measure $d^*$ and the expert occupancy measure $d^E$. To solve this problem, however, the authors propose a practical implementation that completely deviates from the DICE formalism, that uses an in-sample learning offline RL framework like SQL. There are very large gaps between the theoretical derivation and the practical algorithm. Lastly, I think the paper shares quite a lot of similarities with RGM [1] from high-level ideas to problem formulation, but never mentioned RGM in the paper. This actually makes me wonder whether the proposed method is essentially performing reward correction or constraint learning as claimed by the authors. Please see the following strengths and weaknesses for detailed comments.

**Strengths:**

- Constraint learning in the offline setting is a meaningful problem and worth investigating.
- The paper is easy to read and well organized.
- Relative comprehensive experiments.

**Weaknesses:**

- The biggest concern I have is the way this paper models cost values. It approximates cost value as $\lambda_d (D_f(d\| d^E)-\epsilon)$. This is very problematic since being sub-optimal does not necessarily mean it is unsafe or a cost violation. Matching with expert occupancy measures is essentially doing some sort of inverse RL on rewards rather than conventional constraint learning.
- The model construction is highly similar to RGM [1], but it is not even mentioned in the paper. RGM considers a reward correction problem using a similar DICE-based framework. Both the proposed IDVE and RGM use an expert dataset and a large unknown/sub-optimal quality dataset as inputs. Both methods model the problem as similar bi-level optimization formulations: at one level minimize $D_f(d^*\| d^E)$ and on the other level solve an offline RL problem. The only difference is that RGM learns a reward correction term and this paper learns a cost value function $V^c$. Given the similar high-level formulation, I suspect the proposed method is essentially doing some sort of reward correction rather than constraint learning.
- There are many designs in this paper that look not very principled. For example, although the problem is formulated in a DICE framework, the authors use techniques from in-sample learning offline RL algorithms to construct the practical algorithm. First of all, the "state-value" $V$ in DICE-based methods is not the common meaning of state-value functions in typical RL problems, they are actually Lagrangian multipliers. If the authors are familiar with the DICE-class of algorithms, they will notice that the $V$ values in DICE algorithms take very different values and in many cases behave differently from those in common RL algorithms. Hence simply learning value functions using in-sample learning algorithms like IQL or SQL and then plugging them back into a DICE formulation is inappropriate. Also, there are lots of tricks and somewhat arbitrary designs in the practical algorithm, like introducing $\lambda$ in Eq.(12), and the treatment in Eq.(15)-(17). By contrast, RGM offers a much cleaner and more rigorous derivation from theory to practical algorithm.

**Questions:**

- Can you justify the difference between the proposed constraint inference as compared to the reward correction in RGM[1]?
- Why not consider learning $V^r$ and $V^c$ using typical DICE-based techniques, like RGM[1] or SMODICE[2]?

**References:**

[1] Li, J., et al. Mind the Gap: Offline Policy Optimization for Imperfect Rewards. ICLR 2023.

[2] Ma, Y., et al. Versatile offline imitation from observations and examples via regularized state-occupancy matching. ICML 2022.

---

> ### Author Response · Authors · 2023-11-18
> **Author response to Reviewer KCBL - Part 1**
>
> Dear Reviewer, your constructive comments are truly appreciated. Having carefully reviewed them, we sincerely hope the following response demonstrates our dedication to addressing and alleviating any raised concerns:
>
> 1. *'The biggest concern I have is the way this paper models cost values. It approximates cost value as $\lambda_dD_f(d||d^E)$. Being sub-optimal does not necessarily mean it is unsafe or a cost violation.Matching with expert occupancy measures is essentially doing some sort of inverse RL on rewards rather than conventional constraint learning.'*
>
> **Response:**    We apologize for the confusion. **We have updated our paper to derive our learning objective from a constraint learning function (see Section 4) instead of the divergence between two distributions**. We hope our new derivation will serve as a better explanation for our algorithm.
>
> In our revised derivation, we adopt an approach to learning a cost function by introducing a penalty for undesired behavior derived from an optimal policy (objective 9). Then we address the forward optimization challenge of the learned cost function through dual optimization (objective 8). To substantiate the effectiveness of our method, we have established the equivalence condition for a feasible inverse constraint learning solution (refer to Proposition 2) and demonstrated the convergence of this learning process within finite CMDP (refer to Proposition 3). In addition to these theoretical proofs, our paper provides an analysis of our method, with both formulas and a concrete example. We illustrate how our approach leads to a learned solution that exhibits sparsity, a critical concern in constraint learning problems (refer to Section 4.1). We believe that this new derivation effectively illustrates how to transform a constraint learning problem into our bi-level optimization formula, thereby increasing the soundness of our algorithm.
>
> 2. *'Why not consider learning $V_c$ and $V_r$ using typical DICE-based techniques, like RGM[1] or SMODICE[2]?'*
>
> **Response:** Thank you for your concerns. While typical DICE-based techniques primarily focus on inverse reinforcement learning (IRL), directly applying existing IRL methods to recover constraints within the CMDP framework is problematic.
>
> An intuitive extension of IRL to address the ICRL problem would involve employing the Lagrange relaxation technique, transforming the CMDP into an unconstrained problem by treating costs as a penalty term on rewards. Nonetheless, this direct application of IRL lacks a principled approach to:
>
> a) **Utilize existing reward information to guide learning**: Given reference rewards, existing IRL methods do not utilize these signals to recover the penalty term.
> They lack a principled way to incorporate these reference reward signals in their algorithm to guide the learning objective. In practice, these rewards are often simply used for initialization ([1]).
>
> b) **Ensure the recovered reward penalties can be treated as feasible constraints**: The physical meaning of the penalty is the Lagrange multiplier times the costs. Since they are highly entangled, extracting the real costs in the CMDP is difficult.
>
> c) **Incorporating prior knowledge**: Most IRL approaches can't incorporate prior knowledge about the cost function, such as the non-negativity of the cost function. However, due to the well-known non-identifiability of rewards, we can't promise the learned reward is the groundtruth combination of reward and penalty term.
>
> d) **Incorporate assumptions about expert behavior**, i.e., its optimality within the feasible policy set. As detailed in our revised paper, our algorithm maintains the learned occupancy, operating at higher rewards compared to expert occupancy for comparison, in order to recognize the constraint (see Proposition 3). This distinction underscores the fundamental differences between ICRL and IRL.
>
>
> [1] Barnes, M., Abueg, M., Lange, O. F., Deeds, M., Trader, J., Molitor, D., ... \& O'Banion, S. (2023). Massively Scalable Inverse Reinforcement Learning in Google Maps. arXiv preprint arXiv:2305.11290.

---

> ### Author Response · Authors · 2023-11-18
> **Author response to Reviewer KCBL - Part 2**
>
> 3. *'The model construction is highly similar to RGM [1], but it is not even mentioned in the paper. RGM considers a reward correction problem using a similar DICE-based framework. Both the proposed IDVE and RGM use an expert dataset and a large unknown/sub-optimal quality dataset as inputs. Both methods model the problem as similar bi-level optimization formulations: at one level minimize $D_f(d|d^E)$ and on the other level solve an offline RL problem. The only difference is that RGM learns a reward correction term and this paper learns a cost value function. Given the similar high-level formulation, I suspect the proposed method is essentially doing some sort of reward correction rather than constraint learning.'*; *'Can you justify the difference between the proposed constraint inference as compared to the reward correction in RGM[1]?'*
>
> **Response:** We appreciate your assistance in bringing this previous work to our attention, and we apologize for our lack of awareness of it prior to your mention. After a careful review, we agree that our initial derivation shares some modeling techniques and intermediate objectives with the RGM, such as the concept of bi-level optimization.
>
> We have updated our paper to derive our learning objective from a constraint learning function (see Section 4), replacing the divergence between two distributions. We hope this new derivation offers a more comprehensive explanation of our algorithm.
>
> Besides, it's important to note that our work differs significantly from the RGM as we consider different problem settings and objectives. These differences create a substantial gap between our methods in the following aspects:
>
> - In the RGM, the authors **do not provide an explicit mathematical model** for an imperfect reward. They propose four settings of imperfect rewards in Figure 1, but **none of these settings** align with the setting of our Inverse-constrained Reinforcement Learning problem.
>
> - The RGM's aim is to outperform traditional Reinforcement Learning/Imitation Learning techniques, focusing on enhancing forward-control performance to maximize rewards. Conversely, we address an inverse inference problem where the objective is to learn the constraint in order to a) recover the safe policy across all states, and b) allow the transfer of the constraint to other environments with the same dynamics. Our primary concern is safety.
>
> - The derivation of the RGM requires training a discriminator between the demonstration data and the policy-generated data. Our algorithm, however, does not necessitate a discriminator network. Instead, we employ a violation action buffer to better model the problem, as elaborated in section 5.3.
>
> - While their algorithm necessitates minimizing the divergence between the learned policy and the expert policy, our algorithm also can achieves this by maximizing the cost of undesirable behavior derived from raw reward and data. However we derive it from an equivalent condition of constraint learning.
>
> - We demonstrate the effects and dynamics of our learning process and use the chi-square divergence as an example. **This is particularly relevant to sparse constraint learning**, a point not discussed in the RGM.
>
>
>
> 4. *For example, although the problem is formulated in a DICE framework, the authors use techniques from in-sample learning offline RL algorithms to construct the practical algorithm. First of all, the "state-value" $V$ in DICE-based methods is not the common meaning of state-value functions in typical RL problems; they are actually Lagrangian multipliers. If the authors are familiar with the DICE-class of algorithms, they will notice that the $V$ values in DICE algorithms take very different values and in many cases behave differently from those in common RL algorithms. Hence simply learning value functions using in-sample learning algorithms like IQL or SQL and then plugging them back into a DICE formulation is inappropriate.*
>
> **Response:** Thanks for raising the concerns. We acknowledge that the "state-value" $V$ in DICE-based methods does not have the common meaning of state-value functions in typical RL problems, and we have added a description of how to derive policy from the learned "Lagrangian multipliers" instead of treating them as the value function in an "in-sample learning algorithm."
> To be more specific, we want to a policy $\pi_\psi(s,a)$ to sample from $d^*(s,a)$ according to optimal solution $V^r$, so we extract the policy by maximizing  $\max_\psi E_{s,a\sim d^*(s,a)}{\log\pi_\psi(s,a)}=\max_\psi E_{s,a\sim d^O(s,a)}{{\omega_f^*(\delta V^r(s,a))\log\pi_\psi(s,a)}}$. Thank you for correcting our mistake.

---

> ### Author Response · Authors · 2023-11-18
> **Author response to Reviewer KCBL - Part 3**
>
> 5. *Also, there are lots of tricks and somewhat arbitrary designs in the practical algorithm, like introducing $\lambda$ in Eq.(12), and the treatment in Eq.(15)-(17). By contrast, RGM offers a much cleaner and more rigorous derivation from theory to practical algorithm.*
>
> **Response:** Sorry for the confusion. We have revised our paper to provide a detailed explanation of the derivation of our optimization process and the introduction of hyperparameters. This information can now be found in sections 4.1 and 5.1. In section 5.1, we introduced temperature hyperparameters to decouple the original hyperparameters, creating a more effective model for addressing the trade-off in offline reinforcement learning. **This trade-off involves balancing reward maximization and conservatism**. Furthermore, in section 4.1, we introduced the idea of sparsity constraint learning, demonstrated it using the chi-square divergence, and showed that it derives naturally from our formula. We believe that our new derivation for the hyperparameters provides clarity on these concepts.

---

> ### Comment · Reviewer_KCBL · 2023-11-21
> **Additional comments**
>
> I've read the authors' rebuttal and read the revised paper, however, I found more problems.
>
> - Objective (3) in the revised version seems to be ill-defined.
>   - In objective (3), $c(s,a)$ impacts the behavior of $d^{\pi}$, however, $c(s,a)$ itself is underdefined and has no coupling with the maximization objective. I don't think by optimizing (2), we can ever learn $c(s,a)$.
>   - Also, assigning a weight value on the visitation distribution as the cost is strange. It is not clear what is the plausible physical meaning for cost $\times$ probability distribution smaller than some threshold (i.e., $d^{\pi}(s,a)\times c(s,a)\leq \epsilon$).
>   - Lastly, the constraint on $c(s,a)$ in Eq.(3) has a trivil condition: $c(s,a)=0$, which makes this constriant meaningless for constraint learning. In its actual implementation, the authors have to update the cost function to ensure $c(s,a)>0$, this poses extra conditions and already deviates from the initial objective (3).
> - Also, $\delta_V^c(s,a)$ in Eq. (7) is abruptly introduced to replace $c(s,a)$ in Eq. (3). There is no theoretical justification for why the cost value can be directly replaced with some term computed by cost value $V^c$.
> - How objective (9) is derived also remains to be unclear. It is directly introduced without sufficient explanation. I didn't find a detailed theoretical derivation of (9) in the revised version, which makes it hard for me to verify its correctness.
> - The paper Sikchi et al. (2023) is used a lot in this paper to support the validity of this paper, but this paper is currently still an arxiv preprint and has not yet been officially accepted in peer-reviewed conferences/journals. Note that the theoretical derivation and the final practical algorithm in Sikchi et al. (2023) also have some small gaps.
> - The last disturbing fact is that, the revised paper presents a drastically different methodology as compared to the original submission. Even the constraint definition is changed from ($\lambda_d(D_f(d\|d^E)-\epsilon)$ in the original version to $d^{\pi}(s,a)c(s,a)\leq \epsilon$ in the revised version. The final practical algorithm also has some slight differences, I think there might be some issues for the authors to still report the old results. I think all these changes actually turn the original submission into a different paper and warrant another round of careful review, as most of the reviewers' previous judgments are made based on the methodology presented in the initial submission.
>
> Given the above concerns, I've decided to keep my score unchanged.

---

> > ### Author Response · Authors · 2023-11-23
> > **Response to Additional comments**
> >
> > Dear Reviewer, thank you for your thoughtful feedback on our work. We appreciate your feedback, which has been instrumental in refining our methodology.
> >
> > 1. *In objective (3), $c(s,a)$ impacts the behavior of $d^{\pi}$, however, $c(s,a)$ itself is underdefined and has no coupling with the maximization objective. I don't think by optimizing (2), we can ever learn $c(s,a)$*; *Lastly, the constraint on $c(s,a)$ in Eq.(3) has a trivil condition: $c(s,a)=0$, which makes this constriant meaningless for constraint learning. In its actual implementation, the authors have to update the cost function to ensure $c(s,a)>0$, this poses extra conditions and already deviates from the initial objective (3).*
> >
> > **Response:** Sorry for the confusion;  We recognize that we omitted a clear statement regarding our intention to derive a cost function, denoted as $c(s, a)$,  such that solving the forward optimization problem (3) will yield a solution identical to expert behavior $d^E$. It's crucial to clarify that the cost learning formula is explicitly presented in objective (7). Furthermore, we want to emphasize that in our CMDP problem formulation outlined in Section 3, the cost function $c(s, a)$ is constrained to the range of [0, $C_{\max}$]
> >
> >
> > 2. *Also, assigning a weight value on the visitation distribution as the cost is strange. It is not clear what is the plausible physical meaning for cost $\times$ probability distribution smaller than some threshold (i.e., $d^{\pi}(s,a)\times c(s,a)\leq \epsilon$).*
> >
> > **Response:** Thank you for sharing your concern. It's actually a concept corresponding to the state-wise budget, where we guarantee for each state the cost function times the visiting probability is less than the threshold $\epsilon$. When $\epsilon=0$, we have the hard constraint setting.
> >
> >
> > 3. *Also, $\delta_V^c(s,a)$ in Eq. (7) is abruptly introduced to replace the cost value $c(s,a)$ in Eq. (3). There is no theoretical justification for why the cost value can be directly replaced with some term computed by cost value.*
> >
> >
> > **Response:** Sorry for the confusion. Your comment has led us to reconsider our presentation, and we now explicitly state that this term represents the value of the expert policy with respect to the cost function. We actually meant to use the value of the expert policy with respect to the cost function, i.e., $c(s,a)=Q^c_{\pi^E}(s,a) - E_{P(s'|s,a)}[V^c_{\pi^E}(s')]$ to model the cost function.
> >
> > 4. *How objective (9) is derived also remains to be unclear. It is directly introduced without sufficient explanation. I didn't find a detailed theoretical derivation of (9) in the revised version, which makes it hard for me to verify its correctness.*
> >
> > **Response:** Sorry for the confusion. We actually meant to maximize cost in $d^E$ and minimize cost in $d^*$. We have rewritten objective (9) to provide better insight into our objective.
> >
> > 5. *The paper Sikchi et al. (2023) is used a lot in this paper to support the validity of this paper, but this paper is currently still an arxiv preprint and has not yet been officially accepted in peer-reviewed conferences/journals. Note that the theoretical derivation and the final practical algorithm in Sikchi et al. (2023) also have some small gaps.*
> >
> > **Response:** Thank you for your suggestion. Your suggestion to diversify our citations to support the validity of our paper is well-taken. We have incorporated additional references to strengthen the foundation of our work.
> >
> > Once again, we express our gratitude for your thoughtful and constructive feedback. Thank you for your time.

---

### Official Review · Reviewer_86gQ · 2023-10-28

**Soundness:** 2 fair
**Presentation:** 2 fair
**Contribution:** 2 fair
**Rating:** 3
**Confidence:** 3

**Summary:**

This paper studies the inverse constrained reinforcement learning (ICRL) problem. Previous works in ICRL primarily consider the online setting which allows online interactions with the environment. However, in safety-critical tasks, it is often dangerous to iteratively collect samples in the environment since the data-collecting policies may violate the constraints. To address this issue, this paper focuses on the offline ICRL and proposes an Inverse Dual Values Estimation (IDVE) framework. IDVE derives the dual values functions for both rewards and costs, estimating their values in a bi-level optimization problem. To implement IDVE, this paper introduces several techniques: 1) handling unknown transitions, 2) scaling to continuous environments, and 3) controlling the degree of sparsity regularization. The empirical results show that IDVE can accurately recover the constraints and achieve high returns.

**Strengths:**

1. This paper identifies the safety issues in online ICRL. To address this issue, this paper introduces the offline ICRL problem which is more practical.
2. The paper is well-written and easy to follow, providing clear explanations and detailed descriptions of the proposed method and experimental results.

**Weaknesses:**

The derivation of the proposed framework IDEV and its practical implementation introduces many unreasonable transformations, which makes the algorithm lack soundness.

1. In the first paragraph of Section 4.2, the authors approximate the expert regularizer $\lambda_d (D_f (d||d^E) - \epsilon)$ with $\mathbb{E}_d [\delta^c_V (s, a)]$. However, $\mathbb{E}_d [\delta^c_V (s, a)]$ is not a reasonable approximation of $\lambda_d (D_f (d||d^E) - \epsilon)$. There is even no connection between $\mathbb{E}_d [\delta^c_V (s, a)]$ and $\lambda_d (D_f (d||d^E) - \epsilon)$: the former is a temporal difference term w.r.t the cost function while the latter is a divergence between two distributions.
2. The introduction of the lower-level optimization in Eq.(11) is weird. The authors have replaced the expert regularizer with another term. However, they again add this regularizer in the bi-level optimization.
3. In the last line of Section 5.1, they introduce the approximation $Q^c(s, a)=\mathbb{E}_\{\left(s, a, s^{\prime}\right) \in \mathcal{D}^o}\left[\gamma V\_{\theta^c}^c\left(s^{\prime}\right)\right]$. This approximation is incorrect. The correct one is $Q^c(s, a)= c(s, a) + \gamma \mathbb{E}\_{s^\prime \sim p (\cdot|s, a)} [V^c\_{\theta^c} (s^\prime)]$.




Besides, the experiment setup is a little weird. In particular, the offline dataset in grid-world is collected by random policies, and the offline dataset in MuJoCo is collected by SAC policies. Such offline datasets may contain a large number of unsafe behaviors, contradicting the motivation of this paper. Thus, a more proper choice is to apply safe and conservative policies to collect the offline dataset.

**Questions:**

1. What is the meaning of the threshold $\hat{\epsilon}$ in Definition 1?
2. Typos:
    1. In the last paragraph of Section 3, “The demonstration dataset $\mathcal{D}_{O}$” should be “The demonstration dataset $\mathcal{D}^{O}$”.
    2. The notation of $\sum_{p_{\mathcal{T}}\left(s^{\prime} \mid s, a\right)} \gamma V^r\left(s^{\prime}\right)$ is confusing. The correct one should be $\sum_{s^\prime \in \mathcal{S}} p_{\mathcal{T}}\left(s^{\prime} \mid s, a\right) \gamma V^r\left(s^{\prime}\right)$ or $\mathbb{E}\_{s^\prime \sim p\_{\mathcal{T}}\left(\cdot \mid s, a\right) }  [\gamma V^r\left(s^{\prime}\right)]$.

---

> ### Author Response · Authors · 2023-11-18
> **Author response to Reviewer 86gQ**
>
> Dear Reviewer, we want to express our appreciation for the time and effort you dedicated to evaluating our work. Your feedback has been invaluable. We've taken the time to thoroughly consider your suggestions, and we genuinely hope the responses below will address and alleviate concerns you have expressed.
>
> 1. *'However, is not a reasonable approximation of $\lambda_d(D_f(d||d^E)-\epsilon)$ with $E_d[\delta_V^c(s,a)]$. There is even no connection between $E_d[\delta_V^c(s,a)]$ and $\lambda_d(D_f(d||d^E)-\epsilon)$. The former is a temporal difference term w.r.t the cost function while the latter is a divergence between two distributions. The introduction of the lower-level optimization in Eq.(11) is weird. The authors have replaced the expert regularizer with another term. However, they again add this regularizer in the bi-level optimization.'*
>
> **Response**: We apologize for the confusion. The primary goal of the ICRL problem is learning a cost function, and by minimizing the expected cumulative cost, the imitating agent can reproduce the expert behavior (See definition 1, Section 3). **We have updated our paper to derive our learning objective from a constraint learning function** (see Section 4) instead of the divergence between two distributions. We hope our new derivation will serve as a better explanation for our algorithm.
>
>
> In our revised derivation, we adopt an approach to learning a cost function by introducing a penalty for undesired behavior derived from an optimal policy (objective 9). Then we address the forward optimization challenge of the learned cost function through dual optimization (objective 8). To substantiate the effectiveness of our method, we have established the equivalence condition for a feasible inverse constraint learning solution (refer to Proposition 2) and demonstrated the convergence of this learning process within finite CMDP (refer to Proposition 3).
>
> In addition to these theoretical proofs, our paper provides an analysis of our method, with both formulas and a concrete example. We illustrate how our approach leads to a learned solution that exhibits sparsity, a critical concern in constraint learning problems (refer to Section 4.1). We believe that this new derivation effectively illustrates how to transform a constraint learning problem into our bi-level optimization formula, thereby increasing the soundness of our algorithm.
>
> 2. *'In the last line of Section 5.1, they introduce the approximation $Q^c(s,a)=E_{(s,a,s')\in D^o}[\gamma V^c_{\theta^c}(s')]$. This approximation is incorrect.'*
>
> **Response**: Sorry for this typo. We actually intended to express $Q^c(s,a)=E_{(s,a,s')\in D^o}[c(s,a)+\gamma V^c_{\theta^c}(s')]$, but omitted the term $c(s,a)$. Thank you for bringing this oversight to our attention.
>
> 3. *'Such offline datasets may contain a large number of unsafe behaviors, contradicting the motivation of this paper. Thus, a more proper choice is to apply safe and conservative policies to collect the offline dataset.'*
>
> **Response**:     Thanks for the suggestion! We indeed apply safe and conservative policies to collect expert data. This approach ensures that there is no constraint-violating behavior in the expert data. However, when it comes to non-expert data, it may include risk-seeking or unsafe behaviors with larger rewards and costs than those of experts. The constraint can be inferred from the difference between expert and non-expert data.
>
> A realistic example is the task of autonomous driving. We can collect observational data of driving behavior from real life, **which may contain high-risk behaviors like cutting in line, breaking the speed limit, etc**. Meanwhile, we also have access to human behavior with safe driving habits, which is safe and conservative. Therefore, it's reasonable to assume that our dataset contains unsafe behavior.
>
> If both expert and non-expert data were generated by conservative policies, ICRL algorithm cannot infer constraints. This is because we wouldn't have samples from constraint-violating policies, hindering our ability to infer the boundary between safe and unsafe behaviors.
>
> 4. *'What is the meaning of the threshold $\hat \epsilon$ in Definition 1?'*
>
> **Response**: Sorry for the confusion. $\hat \epsilon$ is the threshold of cumulative cost, and in this article, we will address hard constraints; therefore, $\hat\epsilon=0$.
>
> 5. Typos: *'In the last paragraph of Section 3, “The demonstration dataset $D_O$” should be “The demonstration dataset $D^O$”.'*;*'In the last paragraph of Section 3, “The notation of $\sum_{p_T(s'|s,a)}\gamma V^r(s')$ is confusing.'*
>
> **Response**: We apologize for the typos. They have been corrected in latest version.

---

### Official Review · Reviewer_US7L · 2023-10-30

**Soundness:** 2 fair
**Presentation:** 2 fair
**Contribution:** 3 good
**Rating:** 5
**Confidence:** 3

**Summary:**

The paper addresses the offline safe RL scenario where rewards are observable but costs are hidden. The objective is to derive the cost function from the dataset, akin to the IRL setup. The authors introduce IDVE, built on the DICE family's dual formulation of offline RL. It approximates the cost value function by measuring deviations from expert demonstrations since the cost signal isn't directly observed.

**Strengths:**

- The proposed setting appears to be realistic to me.

- The overall algorithmic designs make sense, although I do have concerns with a few choices.

- Section 6.1 effectively visualizes the recovered constraints and the impact of various components.

**Weaknesses:**

- Related works:
Although IDVE is closely linked to the DICE family, formulated with the distribution correction estimation $w(s, a)$, there is no discussion and referencing to the DICE string of works [e.g., 1, 2]. Inclusion of OptiDICE [3] into discussion is also recommended as it also uses a closed-form solution for the inner maximization, in the DICE framework.

- Experiments:

    - While considering no costs, the comparison between IDVE w/oA and offline RL is somewhat questionable. In scenarios like limited arm halfcheetah and blocked halfcheetah, IDVE w/oA shows notably superior cumulative rewards. The inferiority of offline RL suggests potential issues with the baseline's strength or its implementation. Given that the offline RL's objective solely maximizes returns, one would anticipate it to at least match, if not surpass, IVE w/oA in terms of returns.

    - The annotation for the dashed line in Figure 6 appears to be missing. I would also recommend the authors to plot average cumulative rewards/costs for both $D^E$ and $D^{\neg E}$. It will be helpful for the audiences to better understand the numbers in Table 2.

    - (Continued) The gap in returns between expert and sub-optimal demonstrations, as well as between offline IL and expert demonstrations, is unclear. This ambiguity arises because the environments have been modified, eliminating the availability of standardized D4RL scores for comparison. Therefore, plotting rewards/costs of both $D^E$ and $D^{\neg E}$ would be helpful to improve clarity.


[1] Kostrikov, I., Nachum, O., and Tompson, J. (2019). Imitation learning via off-policy distribution matching. arXiv preprint arXiv:1912.05032.

[2] Nachum, O., Dai, B., Kostrikov, I., Chow, Y., Li, L., and Schuurmans, D. (2019). Algaedice: Policy gradient from arbitrary experience. arXiv preprint arXiv:1912.02074.

[3] Lee, Jongmin, et al. "Optidice: Offline policy optimization via stationary distribution correction estimation." International Conference on Machine Learning. PMLR, 2021.

**Questions:**

- Section 5.2: The choice of $\pi \propto \delta^r-\delta^c$ is somewhat odd. Such a $\pi$ won't optimize the inner equation of Eq (7), given the current value estimations. I wonder why was this chosen over $\pi \propto \exp(\delta^r-\delta^c)$?

- Figure 3:
In offline RL/IL, the value functions are solely asscociated with rewards. How, then, were the constraints derived from these methods?

- Table 5 (Appendix): The varying number of expert transitions for different tasks is appears a bit random to me. Could the authors provide clarity on this decision?

---

> ### Author Response · Authors · 2023-11-18
> **Author response to Reviewer US7L**
>
> Dear Reviewer, thank you for your insightful feedback on our work. We appreciate the time you invested in evaluating our work. We've seriously considered your suggestions, and genuinely hope these responses will address and alleviate your concerns:
> 1. *'there is no discussion and referencing to the DICE string of works.Inclusion of OptiDICE [3] into discussion is also recommended as it also uses a closed-form solution for the inner maximization, in the DICE framework.'*
>
> **Response**: Thank you for your suggestion. We have now included a citation for OptiDICE and indeed, both approaches solve a closed-form solution for the inner maximization in a dual problem.
>
> It's crucial to clarify the distinctions between the DICE (DIstribution Correction Estimation) methods from prior work and our approach. The DICE methods:
>
> - Are specifically designed to handle the forward control problem or imitation learning problem.
> - Solve a single-level optimization problem without considering constraints.
>
> In contrast, our method:
>
> - Is designed to address an inverse constraint learning problem under a Constrained Markov Decision Process (CMDP).
> - Solves a bi-level optimization problem (as shown in Equations 8 and 9) **which involves both updating the constraint and the control policy**.
> 2. *"The inferiority of offline RL suggests potential issues with the baseline's strength or its implementation."*
>
> **Response:** Thank you for raising your concerns. We have reviewed our implementation and made necessary adjustments to the hyperparameters. The experiment has been rerun, and the updated results, presented in Table 2 and Figure 6, have been included. The updated results are marked in blue.
>
> 3. *"The annotation for the dashed line in Figure 6 appears to be missing."*
>
> **Response:** Thanks for pointing this out. We have added the missing annotation to enhance the clarity of Figure 6.
>
> 4. *"I would also recommend the authors to plot average cumulative rewards/costs for both offline dataset and expert offline dataset… Therefore, plotting rewards/costs of both offline dataset would be helpful to improve clarity."*
>
>   **Response:** We agree including this information can improve the clarity of our work. We have included plots of average cumulative rewards/costs in Figure 6, for both the offline dataset and the expert offline dataset in our revised paper.
>
> 5. *"Section 5.2: The choice of $\pi\propto\delta^r-\delta^c$ is somewhat odd. Such a $\pi$ won't optimize the inner equation of Eq (7), given the current value estimations. I wonder why was this chosen over $\pi\propto exp(\delta^r-\delta^c)$?"*
>
> **Response:** We apologize for this confusion. In fact, the choice of $\pi \propto \max(\delta^r - \delta^c,0)$ aligns with the use of chi-squared divergence (Section 4.1). In specific, this is the representation of optimal policy when $D_f$ is implemented by chi-squared divergence. On the other hand, $\pi \propto \exp(\delta^r-\delta^c)$ is optimal in terms of Kullback-Leibler divergence. **These choices are optimized under different f-divergences**. We have ensured this is accurately conveyed in the updated version of our paper in section 5.2.
>
> 6. *"Figure 3: In offline RL/IL, the value functions are solely associated with rewards. How, then, were the constraints derived from these methods?"*
>
> **Response:** Sorry for the confusion. We include the value functions from offline IL to study whether we can represent constraints by utilizing only the reward function. Specifically, this function should assign a low value to the constraint-violating behavior and larger value to others. In the case of offline RL, we include the value functions to better illustrate its behaviors in control.
>
> 7.   *"Table 5 (Appendix): The varying number of expert transitions for different tasks appears a bit random to me. Could the authors provide clarity on this decision?"*
>
> **Response:** We apologize for creating this confusion. Across all the experiments, we control the number of expert trajectories to 100. The varying number of expert transitions is due to its dependence on the episode length in the environment. We have revised our paper to better convey this setting.

---

### Author Response · Authors · 2023-11-18
**Summary of updates**

Dear Reviewers, Area Chairs, and Program Chairs,

We deeply appreciate the valuable feedback and helpful guidance received. In response, we have refined our work, focusing on enhancing the derivation of our method, improving overall clarity, and fine-tuning baseline. We have incorporated these improvements into our revised paper. The modifications are highlighted in blue. Here's a summary of the major changes:

- **Replacing the approximation in derivation**: As suggested by Reviewers 86gQ and KCBL, we have provided a more detailed derivation of our methods from the perspective of the constraint learning problem in section 4. This revision **addresses potential ambiguities that may arise from our initial derivation**, where we approximate the divergence with the cost function. Additionally, we provide equivalent conditions for cost learning and theorems for convergence (Propositions 2 and 3) to validate the soundness of our method.

- **Presentation and Clarity**: In accordance with the valuable suggestions from Reviewer US7L and 86gQ, we have fixed the typos and added legends and information about offline data in the experiment in Figure 6 to enhance its presentation. We have ensured that these modifications substantially enhance the overall clarity of the paper.

- **Fine-tuning of Hyperparameters of baseline**: As suggested by Reviewer US7L, we have checked the implementation of our baseline and fine-tuned the hyperparameters. Our new baseline results are shown in Table 2.

- **Provide the analysis of our methods**: In response to the suggestion from Reviewer KCBL, we have provided the analysis of our methods in section 4.1. This analysis highlights the dynamics within our bi-level optimization problem, serving as evidence to distinguish our methods from inverse reinforcement learning methods such as ValueDICE or reward correction methods like RGM.

- **Providing More Explanation of the Derivation of Hyperparameters**: In response to the suggestion from Reviewer KCBL, we have expanded our introduction to the hyperparameters in section 5.1. We have provided detailed information on how the hyperparameter is defined and explained the insights behind the design of such hyperparameters.

We trust that the revisions will effectively address the concerns raised, and we look forward to getting helpful feedback from the reviewers. We are prepared to actively participate in any ensuing discussions.

---

### Meta-Review · Area_Chair_JtnG · 2023-12-05

**Metareview:**

The submitted paper considers inverse constrained reinforcement learning from offline data. To this end, the authors develop the inverse dual values estimation (IDVE) approach which is a bi-level optimization problem that can be applied to offline datasets. Furthermore, the authors take several steps to make the approach more applicable in practical settings and evaluate it in several environments, showing promising results in some cases.

The paper addresses an important problem and comes with relatively thorough experimental validation. However, the reviewers and the discussion between reviewers and authors highlighted several issues in the presentation of the paper, the derivation of the proposed approach, and the discussion of related work. The authors made several updates to their paper during the rebuttal period addressing many of the reviewers' comments, including a different way for devising the proposed approach. While the reviewers appreciated some of the updates, the changes were so substantial, that not all of them could be carefully assessed within the rebuttal period. This also suggests that the authors need to carefully (re-)consider their latest updates and whether these updates ensure the optimal presentation of their method and accurately reflect the relation to other existing works. Thus I am recommending the rejection of the paper and encourage the authors to carefully revise their latest updates to the paper and to adopt other suggestions made by the reviewers for possible future submissions.

**Justification For Why Not Higher Score:**

The paper needs another round of careful review before it should be accepted.

**Justification For Why Not Lower Score:**

N/A

---

### Decision · Program_Chairs · 2024-01-16

Reject